# Historical contingency in parasite community assembly: Community divergence results from early host exposure to symbionts and ecological drift

Rita L. Grunberg[1]*, Brooklynn N. Joyner[1¤], Charles E. Mitchell[1,2]

1 Department of Biology, University of North Carolina, Chapel Hill, Chapel Hill, NC, United States of America,
2 Environment, Ecology and Energy Program, University of North Carolina, Chapel Hill, Chapel Hill, NC, United States of America

¤ Current address: Department of Parks, Recreation & Tourism Management, North Carolina State University, Raleigh, NC, United States of America
* grunberg@email.unc.edu

**Data Availability Statement:** The data and code that support the findings of this study are available through Zenodo at https://zenodo.org/badge/latestdoi/424588449.

## Abstract

Host individuals are commonly coinfected with multiple parasite species that may interact to shape within-host parasite community structure. In addition to within-host species interactions, parasite communities may also be structured by other processes like dispersal and ecological drift. The timing of dispersal (in particular, the temporal sequence in which parasite species infect a host individual) can alter within-host species interactions, setting the stage for historical contingency by priority effects, but how persistently such effects drive the trajectory of parasite community assembly is unclear, particularly under continued dispersal and ecological drift. We tested the role of species interactions under continued dispersal and ecological drift by simultaneously inoculating individual plants of tall fescue with a factorial combination of three symbionts (two foliar fungal parasites and a mutualistic endophyte), then deploying the plants in the field and tracking parasite communities as they assembled within host individuals. In the field, hosts were exposed to continued dispersal from a common pool of parasites, which should promote convergence in the structure of within-host parasite communities. Yet, analysis of parasite community trajectories found no signal of convergence. Instead, parasite community trajectories generally diverged from each other, and the magnitude of divergence depended on the initial composition of symbionts within each host, indicating historical contingency. Early in assembly, parasite communities also showed evidence of drift, revealing another source of among-host divergence in parasite community structure. Overall, these results show that both historical contingency and ecological drift contributed to divergence in parasite community assembly within hosts.

**Funding:** This work was supported by the NSF-USDA joint program in Ecology and Evolution of Infectious Diseases award to CEM (USDA-NIFA AFRI grant 2016-67013-25762).

**Competing interests:** The authors have declared that no competing interests exist.

## Introduction

Most parasite infections occur within the context of a diverse community of parasites, and these parasite coinfections can drive disease dynamics across biological scales [1, 2]. For example, differences in disease burden/severity within host individuals and transmission among host individuals are linked to within-host parasite community structure [1–5]. Parasite communities are also spatially and temporally diverse [6–9]. Variation in parasite community structure across space and time can be driven by host traits and local environmental factors [10–12], as well as interactions among parasite species [1, 3, 13]. However, the role of within-host interactions in driving parasite community assembly has rarely been studied in experiments that allow for natural transmission. Specifically, it is less clear as to what degree the initial composition of parasites within a host generates variation in parasite community structure over time and how this may interact with other processes that structure communities [14, 15].

Conceptual frameworks from community ecology can provide a useful tool to understand the drivers of parasite community structure [16]. One organizational scheme ecologists use to identify community outcomes focuses on four core processes: dispersal, ecological selection, ecological drift, and speciation [17]. As speciation is of greater importance on longer time scales than our study, we do not consider it further. Dispersal rates differ considerably among parasite species, generating variation in the timing and sequence in which multiple parasite species infect host individuals. Further, successful infection by a parasite is subject to ecological selection by the local environment, which includes both abiotic factors and species interactions. Parasite community structure can also be driven by stochastic demographic processes through ecological drift, but the relative importance of drift for parasite communities is less known [18]. Ultimately, these processes together generate variation in parasite community structure among hosts but are not readily evaluated simultaneously.

The initial species composition of parasites within a host can lead to historical contingency in community assembly [19, 20]. This could occur when there are differences in dispersal that generate variation in community composition that in turn alter species interactions [20, 21]. Such scenarios are likely to occur with parasite species that have a high degree of niche overlap and/or modify the host environment in ways that facilitate or impede infections by other species [22]. For example, the first parasite to infect a host can stimulate host immune responses that alter host susceptibility to subsequent parasite infections [23, 24] and modify the outcome of species interactions [3, 25]. Historical contingency has been demonstrated in many host-parasite systems [1, 8, 13, 26, 27] and can shape the distribution [1] and abundance [28] of parasites across hosts. However, initial variation in community composition may not have lasting effects on community structure or it may lead to transient community states that ultimately converge [20], so it is important to consider whether parasite community assembly is largely deterministic (i.e., independent of history) or shows evidence of historical contingency.

Interactions among parasites can also be related to coinfections with other symbiont species (i.e., host-associated microbes, which include parasites, commensals, and mutualists). Parasites commonly interact with a diverse community of symbionts that can modify parasite transmission and infection severity [29]. Mutualistic symbionts are ubiquitous across hosts and can confer disease resistance by stimulating immune pathways associated with defense against certain parasites [30–32] that impede parasite growth [33] and reproduction [34, 35]. However, the importance of mutualists in mediating parasite community structure remains less clear because their overall effects on host susceptibility or parasite growth rates are hard to predict across parasite species within a community. For example, some subsets of parasite species may respond directly to specific immune pathways elicited by a mutualist, but other parasite species may respond indirectly when mutualists modify within host interactions [36, 37]. As responses

of parasites to other symbionts may be species-specific, further studies are required to understand how the effects of symbionts on parasite community assembly depend on the composition of the symbiont community.

Although the role of species interactions in shaping parasite diversity has been explored, relatively little is known about the importance of drift in generating variation in parasite communities [18]. Nonetheless, drift can be an important driver of community patterns in free-living systems [38, 39], and considering its effects, along with deterministic processes, is critical for community ecology [17]. Stochastic fluctuations of the relative abundance of parasite species can occur and have greater impacts on community structure when population sizes are small [17], suggesting that drift is important during the initial stages of infection [18]. Consequently, it is important to consider how drift can influence among-host variation in parasite community structure and how this relationship may change as parasite communities assemble.

To assess the role of ecological selection, dispersal, and ecological drift in driving parasite community assembly, we asked 1) whether different initial parasite community compositions (i.e., simulating variation in initial dispersal) lead to parasite community assembly that is deterministic (governed by ecological selection) or that is subject to historical contingency and 2) how parasite community assembly is influenced by coinfection with a mutualistic endophyte (i.e., additional species interactions and ecological selection). We analyzed parasite species-level responses to the initial symbiont community by quantifying disease progression of each parasite species to assess differences in parasite growth. At the community level, we analyzed trajectories of the parasite communities to test for divergence in community assembly and total temporal change in communities to evaluate how multiple processes (e.g., drift and selection) change parasite communities. In addition, we assessed 3) the role of ecological drift in generating variation in parasite communities among host individuals by quantifying within group variation in parasite community structure over time. We inoculated individual plants of tall fescue (*Lolium arundinaceum*) in the lab with a factorial combination of three fungal symbionts (two parasites, one mutualistic endophyte). Each parasite inoculation treatment, whether it was one or both parasite species together, occurred simultaneously, to represent a different exposure history for an individual host: initial single infection or coinfection. After the inoculations, we deployed the hosts in the field and then conducted longitudinal disease surveys in the field to track parasite community assembly.

In the field, multiple processes drive parasite community assembly [15, 40]. Early in community assembly, stochastic demographic processes could lead to ecological drift, resulting in divergence in community structure among hosts. Additionally, the experimentally imposed differences in initial parasite community structure could result in historically contingent parasite community assembly, leading to persistent differences or even divergence in community structure. On the other hand, as all hosts were exposed to a common pool of parasites and environmental conditions in the field, their parasite communities might converge towards a similar community state. Our goal was to evaluate how these forces, favoring either community convergence or divergence, worked together to shape the assembly of parasite communities within individual hosts.

## Methods

### Study system

Tall fescue in the Piedmont region of North Carolina is commonly infected with multiple species of foliar fungal parasites, making it a useful study system for understanding parasite community assembly. In this study system, parasite epidemics are seasonal: *Colletotrichum cereale*

epidemics begin in spring, *Rhizoctonia solani* epidemics begin in summer, then both decline in fall [1]. Just as host populations are typically infected by these parasites sequentially, so too are host individuals. While host individuals are typically infected first by *C. cereale*, being infected first by *R. solani* reduces host risk of becoming infected by *C. cereale* [1]. This suggests a role of historical contingency by priority effects within host individuals.

Tall fescue is also commonly infected with *Epichloë coenophiala*, (~ 85% prevalence in our study system, B. N. Joyner unpublished data) a systemic fungal endophyte that is transmitted vertically by seed [1]. *Epichloë coenophiala* is considered a mutualist of tall fescue with defensive properties against herbivory [35, 41]. Its effects on disease are less studied and may be parasite species specific, for example, *E. coenophiala* is proposed to facilitate biotrophs and inhibit necrotroph parasites [30]. In addition, *E. coenophiala*, may modify within-host interactions between *R. solani* and *C. cereale* [36]. Together, these past results suggest that understanding parasite community assembly in this system may benefit from experimentally manipulating the composition of symbionts within host individuals, then tracking assembly under field conditions.

## Experiment

Epidemics of C. *cereale* and *R. solani* within the host population are seasonal in our system, but at the individual host-level exposure to both parasite species can occur throughout the growing season. Tall fescue is exposed to C. *cereale* spores throughout the growing season and exposure to R. *solani* is also year-round since it lives in the soil (though active infections occur under certain environmental conditions later in the growing season). To experimentally manipulate the exposure of symbionts within a host, we inoculated individual plants of tall fescue with a factorial combination of three symbionts (2x2x2 design): two foliar fungal parasites, *C. cereale* and *R. solani*, and a fungal endophyte, *E. coenophiala*. The inoculated parasite species *C. cereale* and *R. solani* cause the diseases anthracnose and brown patch, respectively.

**Symbiont inoculations.** In total, we factorially inoculated 136 plants with combinations of three symbionts: *C. cereale* (parasite), *R. solani* (parasite), and *E. coenophiala* (mutualist) (Table 1). *Epichloë coenophiala* is transmitted from infected seed, so we used *E. coenophiala* -free and *E. coenophiala* -infected seeds of the same cultivar KY-31. The *E. coenophiala* -free treatment group comprised 52 plants grown from seeds produced by Tim Phillips at University of Kentucky, and the *E. coenophiala* inoculation group was comprised of 84 plants (a larger

**Table 1. Summary of initial sample sizes and final sample sizes of plants used in analyses after accounting for *Epichloë coenophiala* infection and plant mortality in the field.**

| Inoculation treatment | N (initial) | N (*Epichloë*-positive) | N (mortality) | N (analyzed) |
|---|---|---|---|---|
| No symbiont | 12 | 0 | 1 | 11 |
| Col | 14 | 0 | 0 | 14 |
| Col + Rhiz | 13 | 0 | 0 | 13 |
| Rhiz | 13 | 0 | 1 | 12 |
| Epi | 21 | 8 | 0 | 8 |
| Epi + Col | 21 | 5 | 0 | 0 |
| Epi + Col + Rhiz | 21 | 5 | 0 | 0 |
| Epi + Rhiz | 21 | 1 | 0 | 0 |

N (initial) indicates the number of plants outplanted into the field, N (*Epichloë* infected) is the total number of plants that tested positive for *E. coenophiala* at the end of the experiment, N (mortality) indicates the number of plants that died in the field, and N (analyzed) is the total number of plants included in the analysis after accounting for *E. coenophiala* infections and plant mortality. Shaded in grey are the final 5 treatment groups used in the analyses.

number because we anticipated <100% transmission of the endophyte from seed [36]) grown from seeds produced by the Noble Research Institute in Oklahoma (Table 1). While seed for the two treatment groups were produced by different institutions, all seed was of the KY-31 cultivar, and thus all experimental plants were of the same genetic background. These plants were grown under greenhouse conditions for eight weeks and then exposed to parasite inoculum treatments.

All parasite inoculations were conducted on the oldest living leaf in a designated tiller (i.e., vegetative shoot) within a plant. Our sources of inoculum for *R. solani* and *C. cereale* were derived from cultures of infected tall fescue leaves collected in Duke Forest Teaching and Research Laboratory in Orange County, North Carolina, USA in 2015 by F. Halliday and K. O'Keeffe, respectively [7, 36]. For the *R. solani* inoculation, we placed a mycelium PDA plug at the base of the oldest living leaf to establish infection [36]. Plants that were not inoculated with *R. solani* were mock-inoculated with a blank PDA plug. The inoculation site was then covered with moist cotton and wrapped in aluminum foil and parafilm to maintain a humid environment for parasite growth. After two days, we removed the inoculum covering.

For the *C. cereale* inoculation, we applied a spore solution or PDA broth solution (mock) on the oldest living leaf [42]. We grew C. *cereale* on PDA plates, then flooded the plates with water and scraped them to produce a mycelial and spore solution with a spore density of $10^6$ conidiospores/mL water. Plants were inoculated with C. *cereale* by brushing 0.5 mL of the spore solution onto the oldest leaf. Plants that were not inoculated with the *C. cereale* were mock-inoculated with a solution of autoclaved water and potato dextrose broth. We then maintained all plants for two days in dew chambers, which consisted of a misted sealed bag kept over each individual plant.

We tested whether the symbiont inoculations affected the lifespan (i.e., number of days surveyed in the field) of the inoculated leaf using an ANOVA. We included symbiont inoculation as a main effect with four treatment levels in the ANOVA. Symbiont inoculations influenced the lifespan of inoculated leaves (ANOVA: $F_{4,53} = 3.894$, $p = 0.008$, $\eta^2 = 0.227$). Post-hoc comparisons indicate that parasite inoculations negatively affected the lifespan of the lab-inoculated leaves relative to the lifespan of leaves from *E. coenophiala*-infected plants (S1 Fig).

**Field survey description.** After the inoculations, we moved plants into the field and maintained plants in pots placed into individual holes within a 1.5 x 12 m plot located at Widener Farm in the Duke Forest Teaching and Research Laboratory in Orange Co., North Carolina, USA. This plot was fenced to exclude deer, with an additional mesh barrier to exclude small mammals. Within the plot, there were three rows of 47 holes each and these holes were arranged 0.25 m apart along a 12m length of the plot. Potted plants were placed into holes so that their base was at approximately the same level as that of the resident vegetation. Plant locations were randomized within the plot and each plant was randomly relocated to a new hole every week to homogenize parasite exposure. The plot contained a dense cover of resident tall fescue (~ 80% cover within the plot) to act as a source of parasite inoculum for the experimental plants.

Plants were transferred to the field on 19-Sept-2019, and disease surveys occurred from 19-Sept-2019 to 31-Oct-2019. We conducted our experiment during this time to coincide with several parasite epidemics in our system [1], so hosts would be exposed to several parasite species in the field. We surveyed plants on the first day of deployment into the field and then twice per week for six weeks (N = 13 surveys total, Fig 1). Tall fescue is a long-lived perennial plant; the timescale of our disease survey reflects the lifespan of tillers (weeks to months), which represent our ascribed observational unit within a plant. For each disease survey, we assessed foliar fungal disease symptoms (lesions and pustules) visually. Disease symptoms are an epidemiologically relevant measure of parasite infection because, transmission of our

**Fig 1. Conceptual diagram of key experimental steps, data processing, and data analysis.**

parasites from an infected leaf to another leaf requires production of a lesion or pustule, as these are the source of parasite propagules. Disease symptoms of each parasite species were characterized at the leaf level since infections are localized within an individual leaf [1].

As well as brown patch (*R. solani*; above) and anthracnose (*C. cereale*; above), experimental plants became naturally infected by two other foliar diseases: crown rust (caused by *Puccinia coronata*), and gray leaf spot (caused by *Pyricularia grisea*) (Fig 2). Hereafter, rather than the disease symptoms, we refer to these causal parasite species, reflecting this study's aim to understand the consequences of interactions among symbiont species.

We recorded parasite infection severity (i.e., % area of leaf damaged by a parasite species) by visually identifying lesions on all leaves in the inoculated tiller of a plant ([43], protocol 4.14). In one plant, an inoculated tiller died in the field, so a new daughter tiller was selected for the remaining longitudinal surveys. All leaves in a tiller were surveyed longitudinally for parasites. We identified previously surveyed leaves based on their vertical order within a plant. On the final disease survey, we harvested plants to confirm *E. coenophiala* infection status via immunoblot assay (Agrinostics Ltd. Co, Watkinsville, GA, USA).

## Analysis

The overarching aim of our study was to quantify changes in parasite community structure in response to the host's initial symbiont inoculation. We calculated plant-level infection severity (i.e., parasite infection severity (% leaf area damaged) averaged across leaves within a tracked tiller) for analyses (Fig 1). We used plant-level severity because, even though inoculations were conducted on a single leaf and foliar fungal infections are localized within a leaf [25], *E. coenophiala* infections and plant immune hormones that are hypothesized to mediate parasite within host interactions are not localized within a leaf [44, 45]. Thus, infections of leaves within a plant are not expected to be independent.

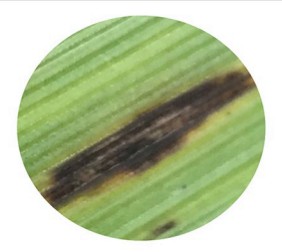

# *Colletotrichum cereale*

Disease: anthracnose
Feeding type: hemibiotroph
Transmission: spores spread by rain splash

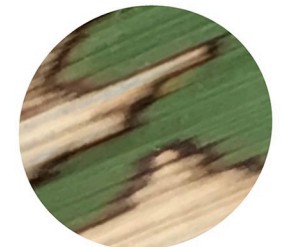

## *Rhizoctonia solani*

Disease: brown patch
Feeding type: necrotroph
Transmission: soil, hyphae and sclerotia

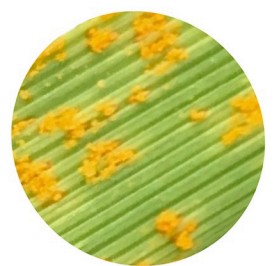

## *Puccinia coronata*

Disease: Crown rust
Feeding type: biotroph
Transmission: air borne spores

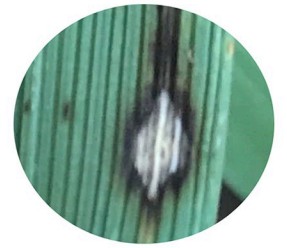

## *Pyricularia grisea*

Disease: grey leaf spot
Feeding type: necrotroph
Transmission: spores spread by rain splash

**Fig 2. Graphical table of the four parasite species and pictures of their associated disease symptoms on tall fescue leaves.** Necrotrophic parasites feed on dead host cells, while biotrophs feed on living cells. Hemibiotrophs employ both biotroph and necrotroph feeding strategies throughout their lifecycle. Pictures were taken by B. N. Joyner.

Detection of *E. coenophiala* in plants grown from *E. coenophiala*-infected seed was generally low. Of the 84 plants grown from *E. coenophiala* -infected seed 15 plants tested positive for *E. coenophiala* at the end of the experiment. None of the *E. coenophiala*-free plants tested positive for the endophyte. Of the four *E. coenophiala* treatment groups, we only retain data from the *E. coenophiala*-only treatment group as to retain statistical power for analyses (n = 8 *E. coenophiala*-only infected plants, 48% positivity, Table 1), and could not test for interactive effects of *E. coenophiala* and parasite inoculations due to low establishment or detection of the *E. coenophiala* within those treatment groups (>5% positivity).

Our analyses consider the effects of inoculation by three symbionts and co-inoculation by the two parasites yielding 5 treatment groups (Table 1). To differentiate between inoculation

treatments and parasite infections detected in the field, we refer to each treatment groups as follows: No symbiont (mock inoculation) Col (*C. cereale* inoculation*)*, Rhiz *(R. solani* inoculation*)*, Col + Rhiz *(*simultaneous *C. cereale* and *R. solani* inoculation, and Epi *(E. coenophiala* positive plants*)*. Single species parasite inoculations also received mock inoculation of the other fungal parasite species. In total, 744 leaves from 136 plants from the original 8 treatment groups were surveyed longitudinally. After accounting for *E. coenophiala* infection within the *E. coenophiala* treatment group and leaves that never became infected, we retained data on 176 infected leaves from 58 plants for analysis (S1 Table).

## Disease progression within hosts

Within host priority effects can have consequences for selection by modifying parasite growth rates. In our system, parasite growth rates are directly linked to changes in parasite lesion size and infection severity over time, hereafter referred to as disease progression. Disease progression was measured by calculating the area under the disease progress stairs (AUDPS) [46] of plant-level infection severity over time in each host. The AUDPS was calculated using the 'agricolae' R package and provides an advantage as it gives a better estimate for the first and final disease measures. We modelled the effects of symbiont inoculation on disease progression across parasite species using a MANOVA. Symbiont inoculation included 4 treatment levels: *E. coenophiala*, *C. cereale*, *R. solani*, and parasite co-inoculation. The response vector of the MANOVA included the AUDPS of *C.cereale*, *R. solani*, and *P. coronata*. *Pyricularia grisea* was excluded from this analysis since it was less prevalent among hosts. *Pyricularia grisea* infected 5% of the plants, while *C. cereale*, *R. solani*, and *P. coronata* more frequently infected plants, with a 92, 47, and 60% prevalence, respectively, so excluding *P. grisea a* provided more power to detect species level effects. The disease progress, AUDPS, was log+1 transformed to minimize heteroscedasticity. A significant response in the MANOVA would indicate disease progression differed between inoculation groups, and if this occurred, we then determined which parasite species contributed to differences in disease progression by conducting univariate ANOVAs for each species. ANOVAs were implemented in the 'car' package with type 3 tests. As a measure of effect size for ANOVAs we reported $\eta^2$, which is the proportion of variance explained by the inoculation treatment in the statistical model. Differences between inoculation treatment groups were then evaluated based on Tukey's HSD post-hoc comparisons.

## Parasite community-level impacts

At the community level, we analyzed parasite community trajectories following the trajectory framework of De Cáceres et al. [47]. This framework focuses on geometric analyses that compare community trajectories, which are represented in multivariate space. We established patterns of community dissimilarity by generating ordinations using Principal Coordinates Analysis (PCoA) on a Bray-Curtis dissimilarity matrix that used parasite infection severity at the plant-level in the 'vegan' R package. On the first disease survey, all hosts showed no visual disease symptoms, which was expected, so to retain these "no parasite community" observations in these analyses we included a dummy variable in the parasite community matrix. The inclusion of the dummy variable causes hosts with no observed parasite community to cluster together, so all hosts started at the same community state and thus had the same coordinates. The parasite matrix contained 4 fungal parasite species and the dummy variable. Because parasite infection severity varied considerably across parasite species (e.g., maximum observed infection severity was 90% of a leaf's area infected by *R. solani*, 50% for *C. cereale*, 3% for *P. grisea*, and 2% for *P. coronata*), we standardized infection severity based on each species

maximum to facilitate a more uniform weighting across parasite species. The endophyte *E. coenophiala* was not included in the parasite community matrix.

### Temporal change in parasite community structure

Temporal changes in parasite community structure can be driven by a combination of selection, dispersal, and ecological drift. By potentially altering parasite growth, infection, and extinction rates, due to selection, the identity of symbionts that initially infect a host may affect how much its parasite community subsequently changes over time. In field conditions that allow for open transmission, dispersal and stochastic demographic processes can also contribute to temporal changes in parasite communities. We measured the extent of temporal change in each parasite community by calculating the cumulative distance moved by the parasite community trajectory. More specifically, for each sequential pair of surveys, we measured the Euclidean distance between the states (i.e., PCoA coordinates) of each community at those two times. The greater the distance between community states at two time points, the more the community has changed over that time interval. As a measure of cumulative temporal change, we then calculated the total trajectory length [47], which is the sum of all distances between sequential community states over the entire sequence of field surveys. We tested whether symbiont inoculations impacted cumulative temporal change in the parasite community using an ANOVA with type 3 tests. A significant effect of symbiont inoculations would indicate that the initial composition of symbionts (i.e., selection) alters the magnitude of change in parasite communities over time. Cumulative temporal change was log10 transformed to minimize heteroscedasticity.

### Parasite community divergence

When communities start in different states, and then are exposed to a common pool of propagules and environmental conditions, they may converge in community structure over time. Thus, parasite communities may converge over time because, after receiving different initial symbiont inoculation treatments, all hosts were exposed to a common pool of parasite propagules for several weeks. To detect signals of community convergence, we first calculated the community centroid of each treatment group in each survey, as a measure of the central tendency of the parasite communities. Second, for each pair of treatment groups, we measured the Euclidean distance between their community centroids at each survey time. Third, we analyzed the relationship of the distance between treatment centroids (i.e., dissimilarity in community states) and time, to evaluate whether community trajectories of the different treatment groups generally became more similar as the communities assembled [47]. To test for convergence or divergence between each pair of treatment groups, we used the non-parametric Mann-Kendall test (R package 'trend'). This test detects monotonic trends in the relationship between dissimilarity (i.e., measured as distance between treatment centroids) and time. In this test, trajectories that are diverging over time have a $\tau$ value greater than 0 (i.e., positive relationship between dissimilarity and time), while trajectories that are converging over time have $\tau$ less than 0 (i.e., negative relationship between dissimilarity and time). $\tau$ values are derived from the sign (positive or negative) of the slopes that describe community divergence; these values do not consider the magnitude of the slopes. To determine the magnitude of trends in community divergence, we also report the sens slope (function 'sens.slope' in the trend package), which is the median of the slope values. Note that the p-values associated with the $\tau$ values in the trend tests also apply to the sens slopes.

## Drift

Finally, we tested whether there was evidence for drift among parasite communities and how the importance of drift changed during the experiment. Specifically, we assessed whether within-group variation increased over time by calculating each parasite community's distance to the group centroid at the time of each survey. An increase in the distance from the centroid over time would indicate that these communities are diverging from a central community state. Such within-group divergence may chiefly represent ecological drift, as our experimental approach sought to eliminate within-group variation in other deterministic factors. We tested for drift among parasite communities using linear mixed effect models. We considered time as a fixed effect and plant ID as a random effect (random intercept only) to account for repeated measurements, and we log10 +1 transformed the distance to the centroid to minimize heteroscedasticity. To test our hypothesis that the importance of drift would decrease over time, we used a piecewise regression model to fit two mixed effect models: one fitting an earlier portion of the community trajectories, and one fitting a later portion. The piecewise approach does this by estimating the break point between the earlier and later mixed models as well as the parameters of each model, with each model's slope being of primary interest (i.e., a change in slope would represent a change in drift). Mixed models were built in 'nlme' package, and the piecewise regression was conducted with the 'segmented' package. All analyses and graphics were produced in R v. 4.0.2 [48].

## Results

### Disease progression within hosts

Across parasite species, disease progression differed between symbiont inoculation treatment groups (MANOVA: $F_{4,53}$ = 2.62, Pillai = 0.496, p = 0.003). The effects of symbiont inoculation on disease progression across all diseases were the product of treatment effects on disease progression of *C. cereale*, *R. solani* and *P. coronata*, based on univariate ANOVAs for each species (Fig 3). *Colletotrichum cereale* and *R. solani* disease progression were impacted by the symbiont inoculation treatments (*C. cereale*: $F_{4,53}$ = 2.92, p = 0.029, $\eta^2$ = 0.181; *R. solani*: $F_{4,53}$ = 2.63, p = 0.044, $\eta^2$ = 0.166), although effects on *P. coronata* were weak and not significant ($F_{4,53}$ = 1.80, p = 0.142, $\eta^2$ = 0.120). Disease progression of *C. cereale* was greater in co-inoculated hosts when compared to hosts inoculated with *C. cereale* only (co-inoculation—Col contrast, p = 0.029, Fig 3A). *Rhizoctonia solani* disease progression was greater when hosts were initially inoculated with *R. solani* relative to hosts that were inoculated with *C. cereale (*Col—Rhiz contrast, p = 0.068, Fig 3B). *Epichloë coenophiala* had no detectable effect on disease progression (all Epi contrasts p > 0.05). Across all symbiont inoculation groups, there was no significant increase in disease progression when compared to the no symbiont exposure group (p > 0.05, Fig 3). Overall, disease progression of all three parasites was generally lowest in hosts inoculated with *C. cereale* only (Fig 3). Effects of parasite co-inoculation on disease progression were species dependent, increasing *C. cereale*, but not *R. solani.*

### Temporal change in parasite community structure

The cumulative temporal change of parasite communities depended on symbiont inoculation treatments (ANOVA, $F_{4,53}$ = 4.946, p = 0.002, $\eta^2$ = 0.272, Fig 3), which explained about 28% of variation among hosts (Fig 4). Plants inoculated with only *C. cereale* harbored parasite communities that changed less over the course of the entire experiment relative to plants inoculated with only *R. solani* (Col–Rhiz contrast, p = 0.007*)* or with both fungal parasites (Col–co-inoculation contrast, p = 0.002). When hosts received no parasite inoculation prior to

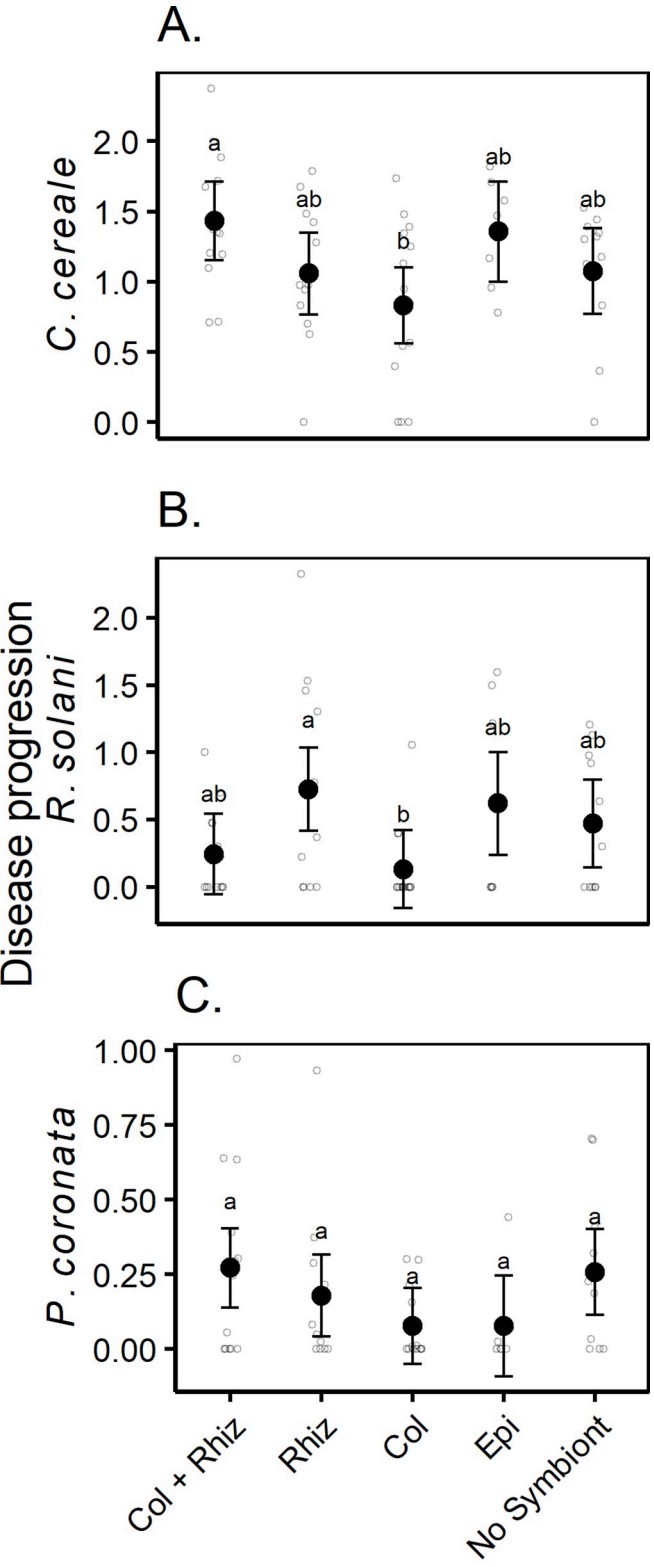

**Fig 3.** Disease progression measured as the area under the disease progress stairs (AUDPS) of A) *C. cereale*, B) *R. solani* and C) *P. coronata* varied in response to individual hosts' initial inoculation with symbionts. No symbiont (mock inoculation) Col (*C. cereale* inoculation), Rhiz *(R. solani* inoculation*)*, Col + Rhiz *(*simultaneous *C. cereale* and *R. solani* inoculation), and Epi *(E. coenophiala* positive plants). Letters denote grouping based on Tukey-HSD post hoc comparisons conducted on disease progression of each parasite species. Filled circles are estimated means and their 95% confidence intervals. Smaller unfilled circles display the raw data that are jittered to show the distribution of the data.

deployment into the field, their temporal change in parasite communities were not statistically distinguishable from other symbiont inoculations (contrasts > 0.05). However, there was weak evidence for differences in temporal variation of parasite communities from *C. cereale* and no symbiont inoculated hosts (Col–no symbiont contrast, p = 0.10).

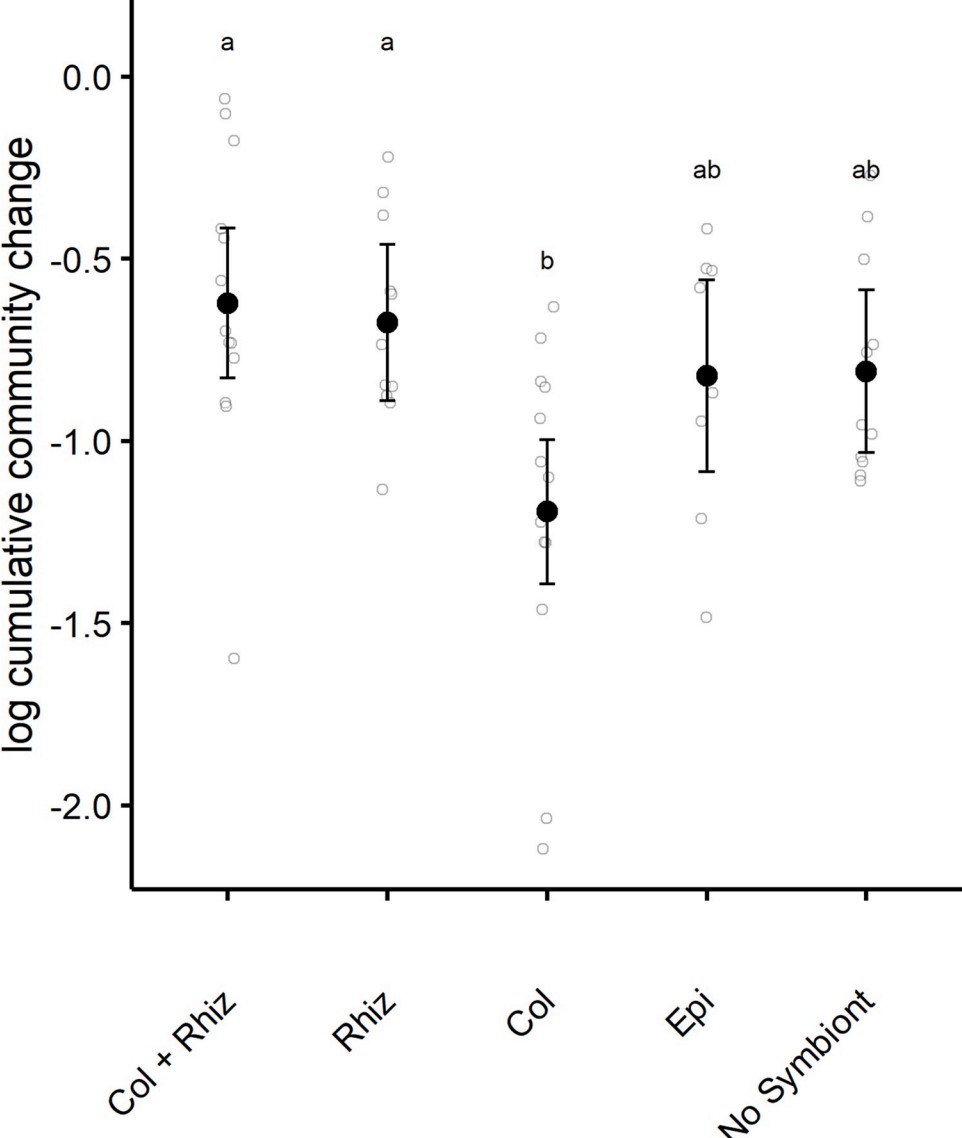

**Fig 4. The cumulative temporal change of parasite communities (i.e., the total distance moved by a parasite community).** Filled circles are the treatment means and their 95% confidence intervals. Letters denote grouping based on Tukey-HSD post hoc comparisons. Smaller unfilled circles display the raw data that are jittered to show the distribution of the data.

## Parasite community divergence

While we expected that exposing all hosts to a common pool of parasite propagules in the field to generate parasite community convergence, trend tests indicated that parasite community trajectories generally diverged between symbiont inoculation treatment groups (Fig 5, S2 Fig). Notably, no treatment groups showed any sign of convergence, as no values of τ were less than zero (τ range: 0.10 to 0.85, S3A Fig). These results indicate persistent changes in parasite community structure over time in response to their initial symbiont inoculations. Specifically, plants that were not inoculated with any symbiont exhibited parasite community trajectories that diverged from other symbiont inoculation treatments (mean τ: 0.59, range: 0.36 to 0.95) except for the parasite co-inoculation treatment, which showed weak evidence of divergence (*C. cereale* and *R. solani*, τ = 0.36, p value = 0.10, S4 Fig). On the other hand, inoculation with the mutualist *E. coenophiala* yielded parasite communities that significantly diverged from the parasite co-inoculated (τ = 0.46, p = 0.03) and no symbiont treated hosts (τ = 0.67, p = 0.001). Parasite community trajectories from hosts co-inoculated with both parasites diverged from those from hosts initially inoculated with *C. cereale* alone (τ = 0.56, p = 0.009).

The magnitude of parasite community divergence, measured by the sens slope, also varied among treatment groups (S3B Fig). Notably, plants co-inoculated with both parasite species, *C. cereale* and *R. solani*, had the greatest mean sens slope value (0.002), indicating that parasite communities in that treatment group diverged the most, on average, from the other treatment groups (S3 Fig). In particular, the greatest rates of divergence occurred between parasite communities from plants co-inoculated with both parasites, *R. solani* and *C. cereale*, and parasite communities from plants inoculated with only *C. cereale* (sens slope = 0.003). The sign of the sens slope is linked to Mann-Kendall trend test, so just as for the τ values, no sens slope values were less than zero, again indicating no convergence.

## Drift

Ecological drift was an important driver of variation in parasite community structure, but only in the beginning of the experiment (Fig 6, S5 Fig). Initially, the distance of the parasite communities to their treatment-group centroid increased over time (slope = 0.0006, 95% CI = (0.0004, 0.0007)), indicating increased variation in parasite community structure within treatment groups over time. Then, piecewise regression identified a break point in the change in distance over time, where after day 21 of the field survey (break point estimate = day 21, 95% CI = (15.5, 28.3)), the distance from the centroid no longer increased (slope = 0.00006, 95% CI = (-0.0001, 0.0002)). Taken together, these results support the prediction that drift is more important earlier in community assembly.

## Discussion

Theory posits that variation in community structure is driven by various processes [17], and we aimed to explore the role of dispersal (i.e., initial community composition), selection (i.e., species interactions), and ecological drift (i.e., among host divergence), in structuring parasite communities, as the roles of the processes are rarely considered simultaneously. Using experimental inoculations and longitudinal field infection data, we found that divergence in parasite community assembly is driven both by prior exposure of hosts to symbionts and ecological drift. Disease progression within hosts inoculated with *C. cereale* only was generally low for all parasite species, and co-inoculation by parasites increased disease progression of some species. At the community level, inoculations also led to community divergence. Parasite communities also diverged considerably among hosts over time, which is indicative of drift. As predicted by theory [18], evidence of drift was only apparent in the beginning of the experiment. Taken

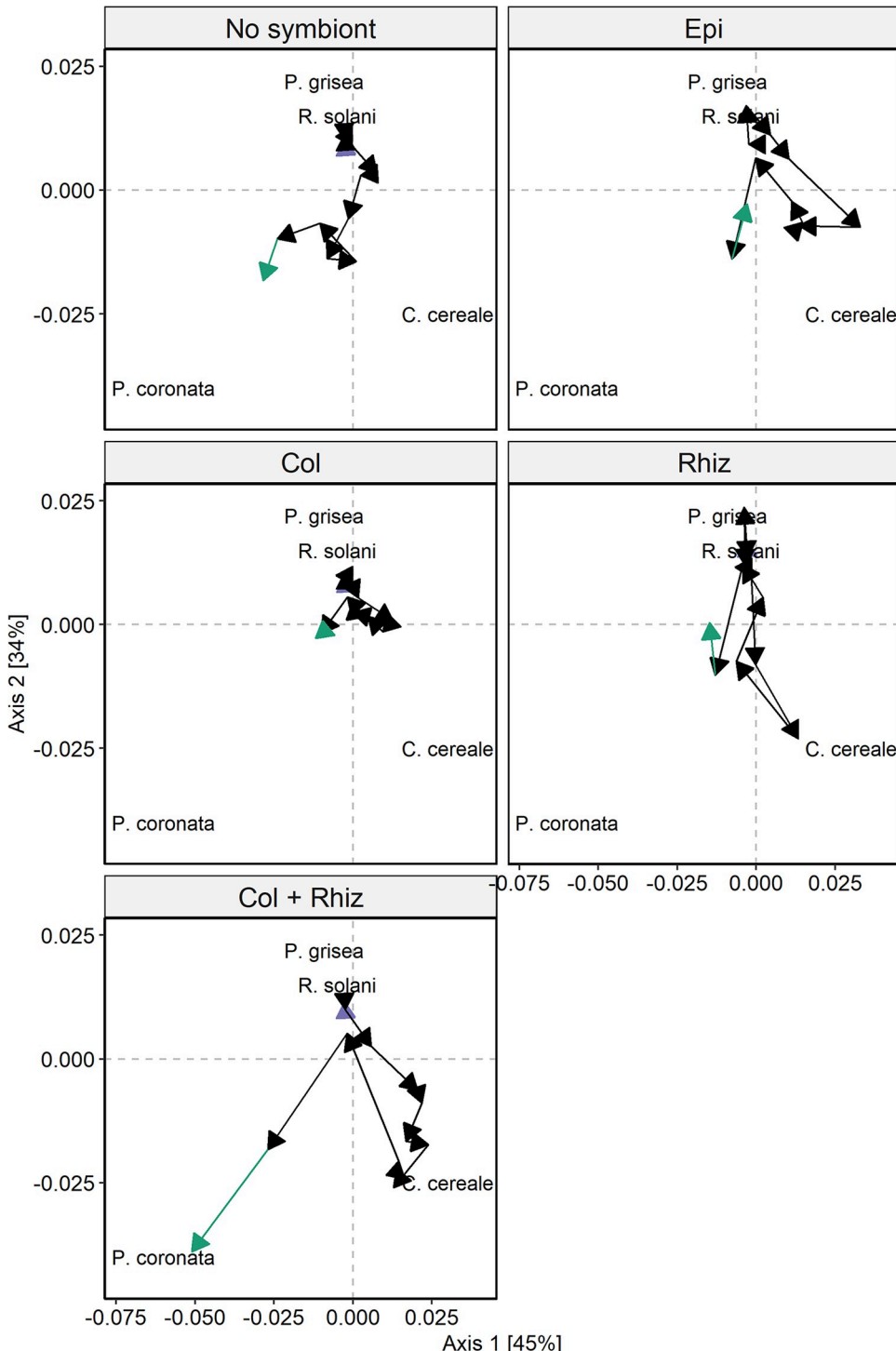

**Fig 5. Divergence of parasite community structure over time.** Each panel shows parasite community patterns for an inoculation treatment group. The arrows show the direction of parasite community assembly over repeated field surveys, and the green arrow indicates the final survey and end of the trajectory. The location of each parasite species name (the first four letters of the genus) indicates the species projection on the PCoA axes.

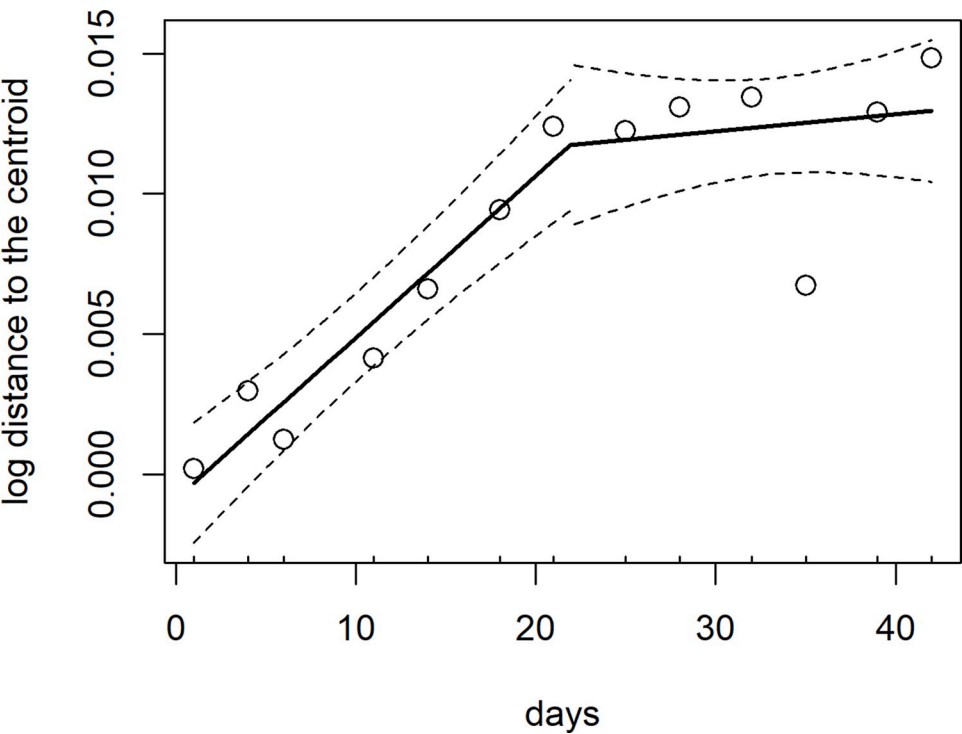

**Fig 6. Early in the experiment, variation in parasite community structure among hosts increased over time, consistent with ecological drift.** The increase in variation of community structure was no longer apparent after day 21 of the field survey, consistent with the prediction that drift would be more important earlier in the experiment. Plotted is the fitted line (solid) and 95% confidence interval (dashed lines) from the piecewise regression mixed model, and each point is the mean variation in community structure across all hosts at a given survey event.

together, these data provide evidence of persistent changes in parasite community structure based on initial variation in parasite community composition and highlight the role of ecological drift as a driver of parasite community divergence.

## Host exposure to symbionts alters disease progression and parasite community trajectories

Even with an identical species pool and environmental conditions, initial differences in species composition could generate alternative community states when community assembly is subject to historical contingency, e.g. resulting from priority effects. Historical contingency by priority effects can alter parasite transmission among hosts [1] and parasite growth rates within hosts [36]. At the community level, differences in parasite growth rates related to historical contingency may result in divergent parasite community trajectories [8, 13]. Notably, parasite communities from hosts initially co-inoculated with both parasites became infected with parasite communities that changed more over time, and generally these communities diverged at a faster rate. The greatest differences in parasite community assembly occurred between hosts co-inoculated with both parasites and *C. cereale* -only inoculated hosts. At one end, inoculation with only *C. cereale* resulted in depauperate communities that did not change much due to low disease progression by all parasite species. However, when hosts were inoculated with both parasite species, the resulting parasite communities changed more over time, and initial co-inoculation increased disease progression of two parasite species, *C. cereale* and *Puccinia*. Throughout many host-parasite systems, coinfections can change the outcome of parasite

interactions [1, 28, 36, 49] and influence parasite community structure [13, 50]. Our data shows that historical contingency due to initial co-inoculation by two parasite species have long-lasting effects on parasite community assembly.

Initial variation in community composition may not always result in persistent changes in community structure. In our experiment, when hosts were initially infected with a single parasite species, this led to changes in disease progression of the other parasite species. Specifically, hosts initially exposed to *C. cereale* had lower disease progression of *R. solani* and hosts initially exposed to *R. solani* had greater disease progression of *R. solani*. However, in this case differences in disease progression did not result in a strong signal of parasite community divergence, and the magnitude of divergence was lowest between the single parasite species inoculated hosts (i.e., *R. solani* v. *C. cereale*). When no strong signals of divergence were found, parasite communities diverged in the beginning and then converged, resulting in no consistent signal. The early signals of divergence and then convergence appear to be related to differences to *Puccinia* infections, which infected hosts later in the experiment. Although single species parasite inoculations altered *R. solani* and *C. cereale* disease progression in ways that resulted in early on divergence, the overall low but consistent levels of *P. coronata* at the end of the experiment led to convergence. Thus, while some parasite communities changed a lot early in the experiment, they ultimately converged towards a similar community state at the end. Convergence at the end of the experiment could be related to differences in selection over time or waning of the parasite epidemics. Specifically, some parasite trajectories tended to end not far from the start (i.e., the initial uninfected community state), which could be related to overall lower infection rates at the end of our experiment. Nonetheless, changes in parasite communities early on can still result in differences in overall host health. So, even when parasite communities end at the same community state, the total path and direction of their infection community trajectory is important to consider. Overall, parasite inoculations altered disease progression of parasite species, which in some cases translated to differences at the community level, with the trajectory of community assembly depending on the identity of parasite species inoculated.

The outcome of parasite species interactions can be modified in the presence of other symbionts of the host, including mutualists [4, 36]. Parasite community trajectories were altered when in the presence of the mutualist *E. coenophiala*. While we did not detect effects of *E. coenophiala* on disease progression, parasite community trajectories from *E. coenophiala*-positive hosts diverged from *R. solani*-inoculated and symbiont-free hosts. The prevalence of *E. coenophiala* in the natural host population is high, so differences in parasite community structure within *E. coenophiala*-positive hosts are relevant for hosts in the field. *Epichloë* interacts with the production of phytohormones associated with plant defense [30, 45]. Changes in certain phytohormones can alter disease in tall fescue by decreasing coinfections and increasing disease severity [25] and may have consequences for parasite community assembly. While we were unable to test for any interactive effects of *E. coenophiala* and initial parasite inoculations, both had important effects on parasite community trajectories. To further understand the role of mutualists in parasite community assembly, future experiments would require treatments involving multiple mutualists taxa since our experiment included a single mutualist species. Therefore, it is still unknown whether other mutualists in this system would have similar effects on parasite community assembly and what traits of the mutualist (i.e., how it interacts with the host immune system) could shape parasite communities.

The importance of a host's initial exposure to symbionts could be sensitive to the time of year we conducted our experiment. Historical contingency in parasite assembly is related to the local pool of parasite species that are able to colonize and interact within a host. For example, if we conducted our field experiment in the early spring rather than the late summer, hosts

would be less likely to develop *R. solani* and *P. coronata* infections because their seasonal epidemics do not start till late summer. As a result, most of our inferences would stem from infections by a single parasite, *C. cereale*, which did differ in disease progression between parasite inoculation groups. However, if effects of initial exposure on disease progression is contingent on the entire parasite community, then we may not see an impact of inoculation on the parasite community at different times in the season. Thus, conducting parasite exposure experiments across different times points that align with other parasite's seasonality would be a valuable next step [51]. In addition to seasonal variation in the parasite species pool, the transmission of many foliar fungal parasites is tightly linked to environmental conditions that also vary over short and long time-scales. Specifically, *R. solani* lives in the environment (soil) year-round but foliar infections occur later in the season under certain environmental conditions. Over longer time scales (e.g., years), such environmental conditions may shift the start and magnitude of parasite epidemics, which could change host exposure rates [1, 52]. In our foliar fungal parasite system, environmental variables (e.g., precipitation and humidity) will play an important role in parasite community assembly and may modify interactions among parasites. To understand the role of historical contingency in parasite community assembly it will be critical to address how sensitive its effects are on communities across gradients of transmission rates during epidemics (variation within a year) and environmental conditions (variation across years).

## Drift contributes to parasite diversity during community assembly

The role of ecological drift in structuring parasite communities has received little attention [15, 18]. In our system, drift was an important structuring force during the early phases of community assembly. During the early phases of infection, parasite communities are expected to be more susceptible to drift due to low population sizes [18]. Similarly, given the high degree of aggregation commonly reported across and within parasite species [53], it is not surprising that most parasite communities exhibit a high degree of among host divergence, because most hosts have low parasite population sizes. Parasite aggregation is largely a product of host heterogeneity i.e., variation in host exposure, behavior, and genetics [54]. We attempted to minimize these sources of host heterogeneity in our study by continually moving plants within the field to homogenize parasite exposure and by using plants from the same cultivar to reduce genetic diversity. Nevertheless, despite our attempt to reduce sources of among-host variation, we detected variation in parasite community structure among hosts, which can be attributed to stochastic demographics processes occurring across parasite species.

The relative contribution of drift will depend on the strength of other processes that structure communities [17]. For example, our study system is not closed and allows for open transmission, so processes like dispersal are still important to consider. At one end, when systems are not dispersal limited, dispersal can homogenize communities and result in convergence over time [55], which we did not detect. However, if species exhibit dispersal limitation, drift and/or selection can be a chief driver of variation in community structure [17, 56, 57]. Further, these relationships between drift and other processes are dynamic through time. As parasite population sizes grow, more deterministic processes (e.g., selection) may contribute more to parasite diversity and thus considering the role of multiple processes throughout community assembly are critical. Our results may not extend to other host-parasite systems that are more complex (e.g., multi-host lifecycle of helminths) or different life history traits. For example, drift may not be as important for parasites that employ more active routes of transmission (i.e., trophic transmission, behavioral modification) [15]. Within some systems, there may be additional within-host dynamics that occur at smaller spatial scales (e.g., host organs or

specific tissues) that may require integrating an entire host landscape to understand the drivers of parasite community assembly. Multiple processes occurring at different spatial scales will drive parasite community assembly in complex systems. This could require delineating different ecological boundaries for parasite communities based on the processes hypothesized to structure the communities [58]. Ultimately, we predict the contribution of drift is an important factor contributing to parasite diversity and the role of stochastic processes should further be considered in the context of parasite community assembly.

## Conclusion

The role of historical contingency and priority effects in structuring parasite communities has been explored in many parasite systems [1, 13, 26, 59]. Our study adds to this knowledge by experimentally evaluating how parasite communities can be structured over time by dispersal, ecological selection, and ecological drift, under field conditions. Importantly, parasite community assembly was subject to historical contingency, so that under similar environmental conditions and exposure rates, parasite communities showed differences in parasite community assembly that were related to their initial exposure to symbionts. Early in community assembly, ecological drift also drove variation in parasite communities and contributed to divergence in parasite community structure over time. More broadly, the relative importance of these ecological processes that structure parasite communities are likely context dependent and may differ based on parasite infection biology and the environmental context. Our study illustrates that considering the role of multiple processes can better our understanding of how parasite communities assembly [3, 60].

## Supporting information

**S1 Fig. Parasite inoculations resulted in a decline in inoculated leaf lifespan (i.e., the numbers of times an inoculated leaf was surveyed, i.e. was still alive) relative to the *Epichloë*-inoculated plants.** Each point indicates an individual leaf that received an inoculation (or mock inoculation) treatment and how many times that inoculated leaf was surveyed in the field. Inoculations were implemented on the oldest living leaf prior to being outplanted in the field. Note that points are jittered to show the distribution of the raw data. Letter denote grouping based on Tukey HSD post-hoc comparisons.
(TIF)

**S2 Fig. Parasite community trajectories differed between symbiont inoculation treatments.** Each panel depicts the parasite community trajectory of each host individual within an inoculation treatment group. Direction of arrows indicate path of each host's parasite community trajectory. Green arrows represent the ending community state for each host.
(TIF)

**S3 Fig.** Parasite community trajectories showed A) trends of divergence between most inoculation groups, and B) the magnitude of divergence between communities was greatest for parasite communities from hosts co-inoculated with both parasite species, *C. cereale* and *R. solani.* Panel A shows the results of pairwise Mann-Kendell trend tests, which test for trends of convergence and divergence between communities. Within cells are Tau values, which indicate whether trajectories show signals of convergence (Tau < 0) or divergence (Tau > 0); the fill of the cells correspond to Tau values, where white cells are 0 and darker blue cells are closer to 1. Panel B shows the values of pairwise sens slopes, which measures the magnitude of divergence. Within cells are the sens slope values, where darker green cells indicate a greater magnitude of divergence between community trajectories. Asterisks in panel A denote significant (p<0.05)

trends of convergence or divergence, and these p-values also correspond to sens slopes in panel B.
(TIF)

**S4 Fig. P-values associated with Mann-Kendell trend tests and sens slopes between parasite community trajectories for each pair of treatment groups.** Asterisks denote significant ($p < 0.05$) trends and sens slopes. Within cells are p-values and the fill of the cells correspond to p-values, where yellow cells are closer to $p = 1$ (non-significant trend) and darker red cells are closer to $p = 0$.
(TIF)

**S5 Fig. Variation in parasite community structure increased as communities developed.** Each panel shows the spread of parasite community structure across all plants for a given sampling event. On the first survey event (panel 1) communities were identical and variation among communities increase throughout assembly.
(TIF)

**S1 Table. Summary infection data of inoculated parasite species in experimental plants.** The # of leaves indicate counts of all leaves that were at any point infected with the target parasite, % prev. is the percent of surveyed leaves that were at any point infected with the target parasite, severity is the average severity across the entire experiment, and total leaves are the number of leaves that were at any point infected by any parasite.
(DOCX)

# Acknowledgments

We thank Dr. Tim Phillips and particularly Dr. Rebecca McCulley for generously providing tall fescue seed. We also thank Tom EX Miller, the Mitchell lab, Peter Morin, and members of the Morin lab for their feedback on this manuscript.

# Author Contributions

**Conceptualization:** Charles E. Mitchell.

**Data curation:** Rita L. Grunberg, Brooklynn N. Joyner.

**Formal analysis:** Rita L. Grunberg.

**Funding acquisition:** Charles E. Mitchell.

**Investigation:** Brooklynn N. Joyner.

**Methodology:** Brooklynn N. Joyner.

**Project administration:** Charles E. Mitchell.

**Resources:** Charles E. Mitchell.

**Supervision:** Charles E. Mitchell.

**Visualization:** Rita L. Grunberg.

**Writing – original draft:** Rita L. Grunberg.

**Writing – review & editing:** Brooklynn N. Joyner, Charles E. Mitchell.

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
