## [Decision Letter · Decision Letter 0]

2 Mar 2023

PONE-D-23-02138Historical contingency in parasite community assembly: community divergence results from early host exposure to symbionts and ecological driftPLOS ONE

Dear Dr. Grunberg,

Thank you for submitting your manuscript to PLOS ONE. After careful consideration, we feel that your manuscript could be accepted after major revisions. Therefore, we invite you to submit a revised version of the manuscript that addresses the points raised during the review process.

We look forward to receiving your revised manuscript.

Kind regards,

Eugenio Llorens

Academic Editor

PLOS ONE

Journal Requirements:

2. In your Methods section, please provide additional information regarding the permits you obtained for the work. Please ensure you have included the full name of the authority that approved the field site access and, if no permits were required, a brief statement explaining why

Reviewers' comments:

Reviewer's Responses to Questions

**Comments to the Author**

1. Is the manuscript technically sound, and do the data support the conclusions?

Reviewer #1: Yes

Reviewer #2: Yes

2. Has the statistical analysis been performed appropriately and rigorously? 

Reviewer #1: Yes

Reviewer #2: Yes

3. Have the authors made all data underlying the findings in their manuscript fully available?

Reviewer #1: Yes

Reviewer #2: Yes

4. Is the manuscript presented in an intelligible fashion and written in standard English?

Reviewer #1: Yes

Reviewer #2: Yes

5. Review Comments to the Author

Reviewer #1: I found the introduction to be well-structured, easy to read, and included the relevant conceptual background for the study. I come from a community ecology background and have expertise in priority effects and historical contingency, but do not work on host-parasite systems; I therefore can’t comment as much on the parasite angle in particular. However, I really enjoyed reading how the authors explained the relevance of dispersal, drift, and historical contingency in the context of host-parasite systems.

The authors performed a study that began in a controlled laboratory setting and then moved to the field and monitored disease progress through time. This is a clever pairing of experimental and observational setups. While I do have some questions about the experimental design, I believe that the conclusions drawn by the authors are supported by their data.

To visualize how parasite communities diverged from one another as well as drifted within a given treatment, it would be helpful to see an overall ordination of all the data on one plat together, maybe from the final timepoint measured. The trajectories in Figure 4 sort of would show this if overlaid on top of each other, but don’t capture the variation within the treatments. Even to include this as a supplementary figure would be helpful in my opinion – to sort of see how the different results dovetail with each other, rather than looking at each process (divergence, drift) separately.

Other specific comments:

From the abstract and last paragraph of the introduction, it was unclear what the treatments comprised – three species are mentioned, but only by digging into the methods can the reader find out whether they were introduced to the host individually, in combination, or both. Also, given that this paper is about historical contingency, it would be good to clarify early on whether introductions were synchronous or sequential in time.

L125-129: Are plants constantly exposed to both R. solani and C. cereale, and other factors determine the timing of epidemics, or does the exposure window differ in time? If plants are usually exposed to the parasites sequentially, then what is the rationale for having all treatments address synchronous exposure scenarios? Is synchronous exposure common in nature, or is it separated in time like the epidemics are?

L125-129: A second question here – given that the epidemics wane in fall, would this affect the degree to which communities might be expected to converge? If the experiment was done in spring or summer would the authors expect different results based on the timing in relation to the epidemics?

L156-160: Are there any potential systematic differences arising from the fact that the Epichloë-laden and Epichloë-free seeds came from different places? How might this affect the experiment?

L235-238: This seems like an interesting result in and of itself – do the authors think that exposure to the other species is causally related to the extremely low Epichloë positivity rate? This could be analyzed statistically.

L297-299: A different way to measure this would be the distance from the original centroid to the final timepoint. I think this would provide complementary information: some trajectories wander around a lot, but ultimately end up not far from the start.

L350 paragraph: The figure actually includes a control treatment as well, where no symbionts were introduced in the lab experiment. It is interesting that the disease progression is never significantly higher for any of the exposure treatments than it is in unexposed plants. The authors should probably address this in the text of the results, not only the figure.

L434-442: This is super interesting! Great finding.

Figure 6 is rotated in the version uploaded with the manuscript.

Reviewer #2: The authors examine the role of three core processes, dispersal, selection and drift, as drivers of community structure of fungal parasites and one mutualist in Lolium arundinaceum. This is pursued by means of monitoring plants experimentally infected with different combinations of fungus species. The study concerns an interesting and little studied topic and the experimental design is appropriate to address the objectives set forth.

The manuscript should be published but not as it stands now. The main issue is that some passages are in the wrong sections or information is repeated in different parts of the text. For instance, the introductory sentences in 287-292 would not be necessary if in the Introduction the three research questions (102-119) were explicitly linked to the five analyses contemplated (i.e. disease progression, community-level impacts, temporal change, divergence and drift). Likewise, contrasting expectations and predictions (l. 398-399 and l. 440-442) with the results obtained should be left to the Discussion. In addition, the legends of Figs. 4-5 reiterate what has been said in the Results section. For example, replace “The trajectory of parasite community structure within hosts inoculated with symbionts generally diverged over time”, with “Divergence of parasite community structure over time.”, and explain the general trend in the main text.

The Conclusion section is not really such, as it mostly summarizes the study. I miss an assessment of the study’s contribution to the current knowledge of the determinants of parasite community structure and how it can guide future research. The authors should also acknowledge some potential limitations that my hinder generalization to other host-parasite systems, such as those formed by macroparasites (i.e. those that not reproduce in the definitive hosts) in vertebrates.

The authors make a distinction between parasite and mutualist species (the latter being Epichloë coenophialia). This species had an effect it determining the community trajectories (506-507), but to which extent the mutualist character of this species accounts for this observation. To prove that mutualists affect the communities in a different way than parasites, one would need to compare the effects of several mutualists species.

Specific comments

It is puzzling that Fig. 1 presents four parasite species when only two have been introduced so far (l. 146). The first mention to this figure should be made in l. 212, thereby becoming Fig. 2.

I know that the habit is deeply rooted, but using generic names as epithets of binomial species names should not be encouraged in scientific publications.

l. 123 Capitalize Piedmont.

l. 211 Insert “naturally” after infected.

l. 270 It should be De Cáceres et al. [48].

l. 320-321 and throughout: replace “tau” or “Tau” with the actual Greek letter τ.

6. PLOS authors have the option to publish the peer review history of their article (what does this mean?). If published, this will include your full peer review and any attached files.

Reviewer #1: No

Reviewer #2: No

---

## [Author Response · Author response to Decision Letter 0]

23 Mar 2023

Reviewer #1: I found the introduction to be well-structured, easy to read, and included the relevant conceptual background for the study. I come from a community ecology background and have expertise in priority effects and historical contingency, but do not work on host-parasite systems; I therefore can’t comment as much on the parasite angle in particular. However, I really enjoyed reading how the authors explained the relevance of dispersal, drift, and historical contingency in the context of host-parasite systems.

The authors performed a study that began in a controlled laboratory setting and then moved to the field and monitored disease progress through time. This is a clever pairing of experimental and observational setups. While I do have some questions about the experimental design, I believe that the conclusions drawn by the authors are supported by their data.

To visualize how parasite communities diverged from one another as well as drifted within a given treatment, it would be helpful to see an overall ordination of all the data on one plat together, maybe from the final timepoint measured. The trajectories in Figure 4 sort of would show this if overlaid on top of each other, but don’t capture the variation within the treatments. Even to include this as a supplementary figure would be helpful in my opinion – to sort of see how the different results dovetail with each other, rather than looking at each process (divergence, drift) separately.

Author: As suggested, we added another figure showing all parasite communities (from all plants) in one figure. Expanding upon this idea, we decided it would be useful to see the spread across all time points, so each panel in this figure is now a survey event. Thank you for the suggestion. This is now Fig S5 in the supplement. 

Other specific comments:

From the abstract and last paragraph of the introduction, it was unclear what the treatments comprised – three species are mentioned, but only by digging into the methods can the reader find out whether they were introduced to the host individually, in combination, or both. Also, given that this paper is about historical contingency, it would be good to clarify early on whether introductions were synchronous or sequential in time.

Author: We clarified the details of the symbiont treatments early in the manuscript: abstract (lines 28-29) and last paragraph of the introduction (lines 115-119).

L125-129: Are plants constantly exposed to both R. solani and C. cereale, and other factors determine the timing of epidemics, or does the exposure window differ in time? If plants are usually exposed to the parasites sequentially, then what is the rationale for having all treatments address synchronous exposure scenarios? Is synchronous exposure common in nature, or is it separated in time like the epidemics are?

Author: This is an interesting question, but difficult to answer because exposure is difficult to observe. First, a plant of tall fescue can live for decades. Perhaps more relevant is a leaf or tiller, which live for weeks to months. Our current understanding is that tall fescue is exposed to spores of C. cereale throughout the growing season, although presumably at varying rates. Tall fescue is also exposed to R. solani propagules throughout the growing season because it lives in the soil year-round, but it is only able to infect and be transmitted between leaves under certain environmental conditions, largely restricting its epidemic to late summer, which is when we conducted our experimental inoculations. So at least in late summer and perhaps throughout the season, we believe that tillers and leaves could be first exposed to either species. However, the experimental design was already complicated, and adding a first/second order of inoculation treatment would have probably made it under-powered and uninterpretable. So, for simplicity, we inoculated both species ~simultaneously. We added a summary of the three sentences preceding this one to the manuscript. See lines: 154-158

L125-129: A second question here – given that the epidemics wane in fall, would this affect the degree to which communities might be expected to converge? If the experiment was done in spring or summer would the authors expect different results based on the timing in relation to the epidemics?

Author: This is an important point, which we now add to the discussion (see lines 548-569). The added text is pasted below:

‘The importance of a host’s initial exposure to symbionts could be sensitive to the time of year we conducted our experiment. Historical contingency in parasite assembly is related to the local pool of parasite species that are able to colonize and interact within a host. For example, if we conducted our field experiment in the early spring rather than the late summer, hosts would be less likely to develop R. solani and P. coronata infections because their seasonal epidemics do not start till late summer. As a result, most of our inferences would stem from infections by a single parasite, C. cereale, which did differ in disease progression between parasite inoculation groups. However, if effects of initial exposure on disease progression is contingent on the entire parasite community, then we may not see an impact of inoculation on the parasite community at different times in the season. Thus, conducting parasite exposure experiments across different times points that align with other parasite’s seasonality would be a valuable next step [53]. In addition to seasonal variation in the parasite species pool, the transmission of many foliar fungal parasites is tightly linked to environmental conditions that also vary over short and long time-scales. Specifically, R. solani lives in the environment (soil) year-round but foliar infections occur later in the season under certain environmental conditions. Over longer time scales (e.g., years), such environmental conditions may shift the start and magnitude of parasite epidemics, which could change host exposure rates [1,54]. In our foliar fungal parasite system, environmental variables (e.g., precipitation and humidity) will play an important role in parasite community assembly and may modify interactions among parasites. To understand the role of historical contingency in parasite community assembly it will be critical to address how sensitive its effects are on communities across gradients of transmission rates during epidemics (variation within a year) and environmental conditions (variation across years).’ 

L156-160: Are there any potential systematic differences arising from the fact that the Epichloë-laden and Epichloë-free seeds came from different places? How might this affect the experiment?

Author: To clarify this point in the manuscript we added the following text: ‘While seed for the two treatment groups were produced by different institutions, all seed was of the KY-31 cultivar, and thus all experimental plants were of the same genetic background.’ See lines: 172-174. 

L235-238: This seems like an interesting result in and of itself – do the authors think that exposure to the other species is causally related to the extremely low Epichloë positivity rate? This could be analyzed statistically.

Author: While this is very interesting, we are wary of interpreting this potential result. Epichloe-tall fescue associations are thought to be highly stable and we don’t know of a mechanism by which parasite inoculation would eliminate Epichloe infection. We consider it more likely that a plant that lacked Epichloe at the end of the experiment may not have had it in the first place (possibly a result of endophyte death during seed storage or some unknown source of mortality). As mentioned, it is also possible that infection with pathogens inhibited Epichloe growth or activity in a way that made them less detectable via immunoblot. Ultimately, we do not have data that can support either scenario. Given this, we did not explore this result in the manuscript. 

L297-299: A different way to measure this would be the distance from the original centroid to the final timepoint. I think this would provide complementary information: some trajectories wander around a lot, but ultimately end up not far from the start.

Author: Thank you for this suggestion. We address this point in the discussion (see lines 521-528), but did not conduct an additional analysis in an attempt to reduce complexity of our study. In our added text, we explain that some communities ended near their initial state (uninfected), which we explain could be due to the waning of the epidemics or variation in selection over time. Further, we also note that any change in the community trajectory early on could result in differences in host health. Text pasted below:

‘Thus, while some parasite communities changed a lot early in the experiment, they ultimately converged towards a similar community state at the end. Convergence at the end of the experiment could be related to differences in selection over time or waning of the parasite epidemics. Specifically, some parasite trajectories tended to end not far from the start (i.e., the initial uninfected community state), which could be related to overall lower infection rates at the end of our experiment. Nonetheless, changes in parasite communities early on can still result in differences in overall host health. So, even when parasite communities end at the same community state, the total path and direction of their infection community trajectory is important to consider.

L350 paragraph: The figure actually includes a control treatment as well, where no symbionts were introduced in the lab experiment. It is interesting that the disease progression is never significantly higher for any of the exposure treatments than it is in unexposed plants. The authors should probably address this in the text of the results, not only the figure.

Author: We emphasized this finding in the results. See line: 391-392. 

L434-442: This is super interesting! Great finding.

Author: Thank you. 

Figure 6 is rotated in the version uploaded with the manuscript.

Author: We apologize the graphic was rotated when it was uploaded. 

Reviewer #2: The authors examine the role of three core processes, dispersal, selection and drift, as drivers of community structure of fungal parasites and one mutualist in Lolium arundinaceum. This is pursued by means of monitoring plants experimentally infected with different combinations of fungus species. The study concerns an interesting and little studied topic and the experimental design is appropriate to address the objectives set forth.

The manuscript should be published but not as it stands now. The main issue is that some passages are in the wrong sections or information is repeated in different parts of the text. For instance, the introductory sentences in 287-292 would not be necessary if in the Introduction the three research questions (102-119) were explicitly linked to the five analyses contemplated (i.e. disease progression, community-level impacts, temporal change, divergence and drift). Likewise, contrasting expectations and predictions (l. 398-399 and l. 440-442) with the results obtained should be left to the Discussion. In addition, the legends of Figs. 4-5 reiterate what has been said in the Results section. For example, replace “The trajectory of parasite community structure within hosts inoculated with symbionts generally diverged over time”, with “Divergence of parasite community structure over time.”, and explain the general trend in the main text.

Author: Thank you for the helpful comments. 

(1) As suggested, we link our analyses to our core aims in the Introduction (see lines 103-115). However, we decided to retain the text explaining the biological rationale for our analyses in the analysis section. The in-depth explanation was intended to help remind/guide the reader of the ecological processes related to our analyses. 

(2) We removed text in the results section that contrast expectations and prediction. 

(3) To reduce redundancy, we edited figure legends as suggested. 

The Conclusion section is not really such, as it mostly summarizes the study. I miss an assessment of the study’s contribution to the current knowledge of the determinants of parasite community structure and how it can guide future research. The authors should also acknowledge some potential limitations that my hinder generalization to other host-parasite systems, such as those formed by macroparasites (i.e. those that not reproduce in the definitive hosts) in vertebrates.

Author: We rewrote the final paragraph so we could clearly identify our contribution to the field and contrast it with other systems. In brief, we state that our results are likely system specific, and the relative importance of these processes will differ based on the parasite-host biology. See lines: 608-619. We also expand upon this suggestion in the drift portion of the Discussion section (lines 590-603) to further explain how some differences in parasite infection biology may drive the relative importance of drift in other systems. Further, we also discuss incorporating other within-host dynamics occurring at smaller spatial scales (organs or tissues in animals) to better understand parasite community assembly. 

The authors make a distinction between parasite and mutualist species (the latter being Epichloë coenophialia). This species had an effect it determining the community trajectories (506-507), but to which extent the mutualist character of this species accounts for this observation. To prove that mutualists affect the communities in a different way than parasites, one would need to compare the effects of several mutualists species.

Author: We agree with this point. In the discussion (lines 543-548), we added the following text: ‘To further understand the role of mutualists in parasite community assembly, future experiments would require treatments involving multiple mutualists taxa since our experiment included a single mutualist species. Therefore, it is still unknown whether other mutualists in this system would have similar effects on parasite community assembly and what traits of the mutualist (i.e., how it interacts with the host immune system) could shape parasite communities.’ 

Specific comments

It is puzzling that Fig. 1 presents four parasite species when only two have been introduced so far (l. 146). The first mention to this figure should be made in l. 212, thereby becoming Fig. 2.

Author: Thank you for pointing this out. We switched the order of Fig 1 and 2. 

I know that the habit is deeply rooted, but using generic names as epithets of binomial species names should not be encouraged in scientific publications.

Author: This was fixed throughout the manuscript. 

l. 123 Capitalize Piedmont.

Author: Fixed. 

l. 211 Insert “naturally” after infected.

Author: Inserted naturally after infected. 

l. 270 It should be De Cáceres et al. [48].

Author: Fixed. 

l. 320-321 and throughout: replace “tau” or “Tau” with the actual Greek letter τ.

Author: Fixed.

---

## [Decision Letter · Decision Letter 1]

12 Apr 2023

PONE-D-23-02138R1Historical contingency in parasite community assembly: community divergence results from early host exposure to symbionts and ecological driftPLOS ONE

Dear Dr. Grunberg,

Thank you for submitting your manuscript to PLOS ONE. The reviewers are satisfied with the comments and corrections. However, reviewer 2 suggested some minor changes, including moving data from supplementary to the main text.  Therefore, we invite you to submit a revised version of the manuscript that addresses the points raised during the review process.

We look forward to receiving your revised manuscript.

Kind regards,

Eugenio Llorens

Academic Editor

PLOS ONE

Journal Requirements:

Reviewers' comments:

Reviewer's Responses to Questions

**Comments to the Author**

1. If the authors have adequately addressed your comments raised in a previous round of review and you feel that this manuscript is now acceptable for publication, you may indicate that here to bypass the “Comments to the Author” section, enter your conflict of interest statement in the “Confidential to Editor” section, and submit your "Accept" recommendation.

Reviewer #1: All comments have been addressed

Reviewer #2: All comments have been addressed

2. Is the manuscript technically sound, and do the data support the conclusions?

Reviewer #1: Yes

Reviewer #2: Yes

3. Has the statistical analysis been performed appropriately and rigorously? 

Reviewer #1: Yes

Reviewer #2: Yes

4. Have the authors made all data underlying the findings in their manuscript fully available?

Reviewer #1: Yes

Reviewer #2: Yes

5. Is the manuscript presented in an intelligible fashion and written in standard English?

Reviewer #1: Yes

Reviewer #2: Yes

6. Review Comments to the Author

Reviewer #1: The authors have done an excellent job responding to the reviewer comments, including adding substantial text to the discussion. The methods are also now more clear. I want to congratulate the authors on this, and I have no further comments or questions for them.

Reviewer #2: As I indicated in my first review, this study concerns an interesting and little studied topic in community ecology. The combination of experimental and field setups is appropriate and I believe provides interesting insight into the assembly of pathogen-mutualist communities in tall fescue.

I am satisfied with the answers and corrections done by the authors. I would highlight in particular the addition of an assessment of the contribution to the field and a discussion of applicability to other host-parasite/mutualist systems as elements that add value to the study. Thus, I have no reservation in recommending the article for publication in PLOS One.

There are some minor issues that the authors should consider:

1. Table S1 contains information that is highly relevant for the reader to grasp the structure of the experimental design. I would move it to the main text.

2. Here and there (e.g., 282-283), C. cereale is written with an additional blank space.

3. 301 – Replace “distance” with “dissimilarity”. The Bray Curtis metric is a dissimilarity, not a distance (see e.g. Roberts 2017, https://doi.org/10.1111/2041-210X.12739, for details).

4. 482 – “As predicted by theory”. Please provide a reference.

7. PLOS authors have the option to publish the peer review history of their article (what does this mean?). If published, this will include your full peer review and any attached files.

Reviewer #1: No

Reviewer #2: No

---

## [Author Response · Author response to Decision Letter 1]

13 Apr 2023

Review Comments to the Author

Reviewer #2: As I indicated in my first review, this study concerns an interesting and little studied topic in community ecology. The combination of experimental and field setups is appropriate and I believe provides interesting insight into the assembly of pathogen-mutualist communities in tall fescue.

I am satisfied with the answers and corrections done by the authors. I would highlight in particular the addition of an assessment of the contribution to the field and a discussion of applicability to other host-parasite/mutualist systems as elements that add value to the study. Thus, I have no reservation in recommending the article for publication in PLOS One.

There are some minor issues that the authors should consider:

1. Table S1 contains information that is highly relevant for the reader to grasp the structure of the experimental design. I would move it to the main text.

Author: We moved Table S1 to the main text (see lines 177). We agree it will help the reader understand the experimental design.

2. Here and there (e.g., 282-283), C. cereale is written with an additional blank space.

Author: Fixed. 

3. 301 – Replace “distance” with “dissimilarity”. The Bray Curtis metric is a dissimilarity, not a distance (see e.g. Roberts 2017, https://doi.org/10.1111/2041-210X.12739, for details).

Author: Fixed. 

4. 482 – “As predicted by theory”. Please provide a reference.

Author: We added the corresponding reference; Seabloom et al (2015). The community ecology of pathogens: Coinfection, coexistence and community composition. Ecology Letters, 18(4), 401–415.

---

## [Editor Report · Decision Letter 2]

16 Apr 2023

Historical contingency in parasite community assembly: community divergence results from early host exposure to symbionts and ecological drift

PONE-D-23-02138R2

Dear Dr. Grunberg,

We’re pleased to inform you that your manuscript has been judged scientifically suitable for publication and will be formally accepted for publication once it meets all outstanding technical requirements.

Kind regards,

Eugenio Llorens

Academic Editor

PLOS ONE

---

## [Editor Report · Acceptance letter]

4 May 2023

PONE-D-23-02138R2 

Historical contingency in parasite community assembly: community divergence results from early host exposure to symbionts and ecological drift 

Dear Dr. Grunberg:

I'm pleased to inform you that your manuscript has been deemed suitable for publication in PLOS ONE. Congratulations! Your manuscript is now with our production department. 

Kind regards, 

on behalf of

Dr. Eugenio Llorens 

Academic Editor

PLOS ONE